# Urban risk factors for human Rift Valley fever virus exposure in Kenya

Keli Nicole Gerken[1]*, Francis Maluki Mutuku[2], Bryson Alberto Ndenga[3], Gladys Adhiambo Agola[3], Eleonora Migliore[1], Eduardo Palacios Fabre[1], Said Malumbo[4], Karren Nyumbile Shaita[3], Izabela Mauricio Rezende[1], A. Desiree LaBeaud[1]

1 Department of Pediatrics, Division of Infectious Diseases, Stanford University School of Medicine, Stanford, California, United States of America, 2 Department of Environment and Health Sciences, Technical University of Mombasa, Mombasa, Kenya, 3 Centre for Global Health Research, Kenya Medical Research Institute, Kisumu, Kenya, 4 Vector Borne Disease Control Unit, Msambweni County Referral Hospital, Kwale, Kenya

* kelingerken@gmail.com

**Data Availability Statement:** Data is available in a repository hosted by Stanford University, and can be accessed by the following link: https://purl.stanford.edu/fp177dq8431.

## Abstract

The Rift Valley fever virus (RVFV) is a zoonotic arbovirus that can also transmit directly to humans from livestock. Previous studies have shown consumption of sick animal products are risk factors for RVFV infection, but it is difficult to disentangle those risk factors from other livestock rearing activities. Urban areas have an increased demand for animal source foods, different vector distributions, and various arboviruses are understood to establish localized urban transmission cycles. Thus far, RVFV is an unevaluated public health risk in urban areas within endemic regions. We tested participants in our ongoing urban cohort study on dengue (DENV) and chikungunya (CHIKV) virus for RVFV exposure and found 1.6% (57/3,560) of individuals in two urban areas of Kenya had anti-RVFV IgG antibodies. 88% (50/57) of RVFV exposed participants also had antibodies to DENV, CHIKV, or both. Although livestock ownership was very low in urban study sites, RVFV exposure was overall significantly associated with seeing goats around the homestead (OR = 2.34 (CI 95%: 1.18–4.69, *p = 0.02*) and in Kisumu, RVFV exposure was associated with consumption of raw milk (OR = 6.28 (CI 95%: 0.94–25.21, *p = 0.02*). In addition, lack of piped water and use of small jugs (15–20 liters) for water was associated with a higher risk of RVFV exposure (OR = 5.36 (CI 95%: 1.23–16.44, *p = 0.01*) and this may contribute to interepidemic vector-borne maintenance of RVFV. We also investigated perception towards human vaccination for RVFV and identified high acceptance (91% (97/105) at our study sites. This study provides baseline evidence to guide future studies investigating the urban potential of RVFV and highlights the unexplored role of animal products in continued spread of RVFV.

## Introduction

The Rift Valley fever virus (RVFV) is a zoonotic arbovirus that can cause severe disease in both humans and animals. Cattle, sheep, goats, and camels are the most affected domestic species, although RVFV is highly adaptable to many mammalian and vector hosts and thus, the disease ecology can vary between locations [1, 2]. Human to human transmission has not been

**Funding:** This study was funded by NIH Fogarty Global Health Equity Scholars Program NIH, D43TW010540 (KG) and NIH, R-01 AI102918 (PI: ADL). The funders had no role in study design, data collection and analysis, decision to publish, or preparation of the manuscript.

**Competing interests:** The authors have declared that no competing interests exist.

documented, except in congenital infections [1, 3]. Instead, RVFV is transmitted to humans by infected vectors or by viral aerosolization from infected livestock tissues and body fluids [4]. During large outbreaks, RVFV causes mass mortality of young livestock and near 100 percent abortion in pregnant animals [1]. As livestock succumb to RVF disease, the humans that rely on them to support their livelihoods attempt to mitigate their losses by assisting in the removal of abortion tissues, slaughtering, or selling sick animals, and sometimes consuming these animal sourced foods (ASFs), all of which put them at high risk of severe disease or even death from RVF disease [5, 6]. Thus, risk of RVFV exposure is highly connected to livestock rearing activities, human behavior, cultural norms, and the ever-changing ecosystems in which susceptible hosts live [7, 8]. Because of the current global distribution of RVFV, the risk of more widespread risk factors such as consumption, and handling of raw milk has not been disentangled from livestock rearing activities. There is also very little field data on the infectious nature of milk which, highlights a mechanism for RVFV transmission without owning or working with livestock [5, 9].

Congested urban areas have an increased demand for animal source foods, different vector distributions driven by land use change, crowding of animals and humans, and significant inequalities within a small geographical area, making them hotspots for many infectious diseases [10, 11]. Although urban invasion of RVFV has yet to occur, various other arboviruses have re-emerged in urban areas, and caused devastating large-scale urban outbreaks [12]. Urban centers of endemic countries, such as Kenya, have all necessary hosts and vectors to support RVFV transmission [2]. Previous RVFV expansion is thought to have been driven by livestock movement [13], and since urban areas do not raise enough livestock to support the higher demand for ASFs, livestock are brought into slaughter from a wide geographical range [10]. Urban areas have also received RVFV infected human patients during outbreak times [14] whose viremias are high enough to infect naive vectors [15], though this has not been shown in a field setting. In addition to multiple introduction points, previous studies in Kenya have shown that the critical *Aedes spp*. capable of vertical transmission and responsible for worldwide outbreaks of DENV and CHIKV are denser and have a preference for urban areas compared to rural areas [1, 12, 16–19]

When considering the geographical range of susceptible vectors and mammalian hosts, most of the world's people and mammals are theoretically at risk of RVFV which makes it a pathogen of concern for most global public health agencies [20]. Historically, major transitions in human lifestyle, such as the rise of agriculture, have resulted in the emergence of new diseases and new disease patterns [21], and for arboviruses, these changes are exacerbated by unpredictable weather patterns driven by climate change are projected to shift the burden from malaria to arboviruses in Africa [8, 22, 23] The current global urbanization trend represents a vulnerable time in zoonotic disease history, and we are unprepared. The current lack of RVFV risk assessments in urban areas of endemic counties puts a significant portion of the population at risk and undermines preventive measures for worldwide RVFV emergence and control. RVFV outbreaks are becoming more sporadic from year-to-year and previously low risk areas of endemic countries are experiencing human outbreaks without any evidence of prior livestock infections which supports a potential low-level circulation of RVFV in livestock [24].

We hypothesize that an urban transmission cycle of RVFV is indeed possible and that the first urban infections may not occur as a detectable outbreak, but rather through cryptic transmission from infected livestock or infected animal products. In this study, we aimed to document the human community burden of RVFV in two areas of Kenya and describe how risk factors may vary in the urban setting compared to what we have collectively learned previously about RVFV studies conducted primarily in rural areas [5, 25–27].

## Materials and methods

This study was conducted at two urban sites in Kenya, Kisumu and Ukunda, from December 2019—April 2021 in collaboration with a large ongoing cohort study on the epidemiology of dengue (DENV) and chikungunya (CHIKV) viruses (NIH, R01 AI102918; PI: ADL). Briefly, for the parent study, a community cohort was recruited from distinct zones within the residential area of two urban centers in Kenya (Fig 1). Households were identified and all people one year and older were consented and enrolled in the study. Understanding of voluntary consent was verified with a checklist and participants agreed to undergo venipuncture for baseline serum IgG antibody testing for anti-DENV, anti-CHIKV and other arboviruses, a complete clinical examination, respond to a questionnaire, and be contacted again for follow-up visits. The structured survey from enrollment included questions on household water source, mosquito prevention measures, and recent travel. Follow-up household visits for the parent study occurred six months after enrollment, and at this time, RVFV questions were incorporated, and blood samples were taken again. These questionnaires have been included as S1 Appendix. A full clinical examination also was recorded at each visit although data was not examined for this RVFV exposure study.

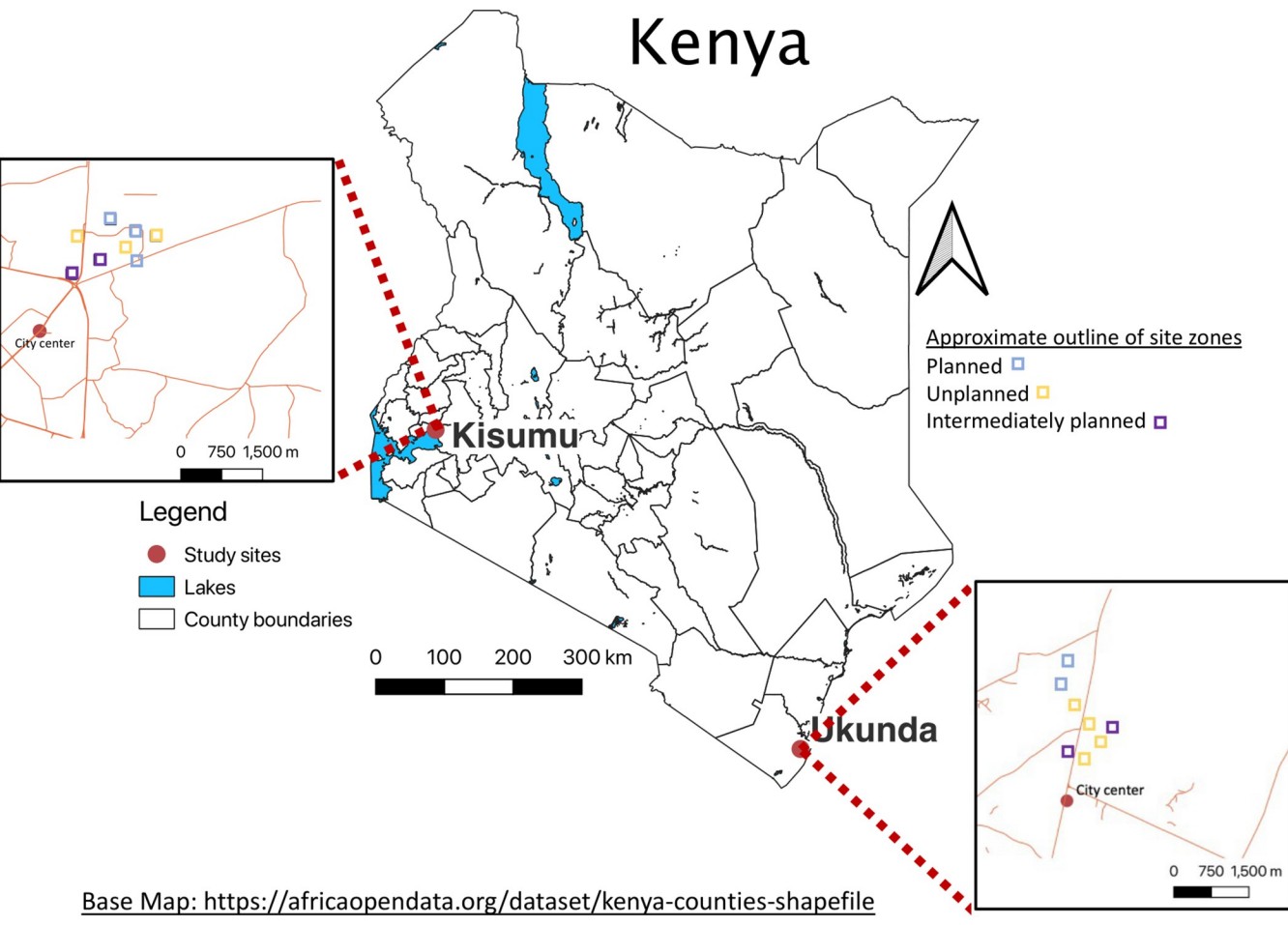

**Fig 1. Map of urban study sites, Kisumu and Ukunda, Kenya.** Base Map: https://africaopendata.org/dataset/kenya-counties-shapefile.

## Study locations

The study sites for this project included all participants from the parent study living in two urban centers of Kenya: Kisumu (west) and Ukunda (coast) (Fig 1). In both urban centers, agriculture, including both crop and livestock production, is one of the main economic activities. At each study site, 200x200 meter zones (Fig 1) were set in the heart of densely populated urban areas to capture a diverse urban population. Zones were assigned letters (A- R) and categorized as having unplanned, unplanned intermediate, and planned housing arrangements which were determined by the accessibility, housing materials, and overall organization.

Kisumu (0˚5' 15.2247800 S, 34˚46' 22.328400 E) is the third largest city in Kenya and a business hub located at the shores of Lake Victoria in western Kenya at an altitude of 1,100 meters above sea level, and a population of 610,082 inhabitants in the 2019 census [28]. Some sections of the city are well-developed (planned) and maintained as opposed to others which are poorly drained (unplanned), where water accumulates and attracts breeding mosquitoes [29]. Kisumu County normally has two rainy seasons a year: short rains between September and November and the long rains between March and May with an average year-round temperature of 23.1˚C ranging from 16˚C to 33˚C. The total precipitation is 966 mm per year with an average humidity of 63% [30]. Studies conducted near to Kisumu in western Kenya between 2010 and 2012 revealed a baseline community seroprevalence of 0.8% and slaughterhouse workers had 2.5% anti-RVFV IgG seropositivity [31].

Ukunda (4˚16' 38.8992" S, 39˚34' 9.0012" E) is an emerging coastal urban center located 30 kilometers south of Mombasa and is the largest city in Kwale County. It is located at an elevation of 24 meters above sea level and is two kilometers from Diani beach which is a frequent tourist destination. It began to expand in the 1970s initially as a residential area for beach resorts workers but has grown into a small commercial center with an estimated population of about 100,000 people and thus represents an area of recent urbanization in Kenya [28]. Rainfall is bimodal with long rains occurring from March to June and short rains from October to December. Total rainfall is on average 1060 mm per year with annual mean temperatures of 28˚C ranging from 23˚C to 34˚C, and average relative humidity of 74%. RVFV transmission in Kwale was first reported in 1961, and Kwale has been involved in 10 national RVFV outbreaks [32]. Recent studies classified Kwale County among the high-risk areas for epizootic RVFV transmission [33], but a low-risk transmission area for humans [25].

## Ethics statement

The study in its entirety was approved by the Institutional Review Board (IRB) at Stanford University (IRB-57869) and the Technical University of Mombasa IRB (TUM ERC EXT/004/2019 (R)). Formal consent was obtained by giving the participant adequate time to read the consent form document, reviewing their questions, and understanding of the consenting process was confirmed with a checklist before signature. For participant under 18 years of age, the parent or guardian of each child consented on their behalf. For this RVFV study, the parents of child participants had previously agreed for their child's sample to be tested for arboviruses when they were enrolled in the parent study. This RVFV study did not have permission to ask additional questions to participants less than 18 years old and therefore, a proxy parent or guardian answered these questions for RVFV-exposed children. The funders had no role in study design, data collection and analysis, decision to publish, or preparation of the manuscript.

## Study design: Stages 1–3

Each stage (1–3) of this study design is outlined in detail below. Risk factors for the full cohort (Stage 1) and the animal exposure questions (Stage 2), were assessed as a cross-sectional study design. Nested case-control data (Stage 3) were collected in series and analyzed separately.

**Stage 1: Baseline demographics and enrollment.** At enrollment in the parent study, each participant underwent venipuncture and received a questionnaire (S1 Appendix). The head of the household responded to the demographics section that included factors such as the family's religion, tribe, water source, roofing type, and ownership of mosquito bed nets. Household level demographic factors were merged with all participants, including children, living in that household. Each individual participant also answered questions regarding their education, recent travel, recent fevers, and mosquito avoidance behaviors. The clinical serum samples tested for RVFV exposure were also simultaneously tested for exposure to DENV and CHIKV and this information has been used to report on arboviral co-exposures.

**Stage 2: Animal-exposure risk factors.** Per our IRB approval, only participants older than 18 years old were asked questions about known RVFV animal husbandry risk factors at their first follow-up visit. This questionnaire was delivered from September 2020-January 2021 and included basic animal exposure questions, and known RVFV risk factors such as livestock ownership, seeing animals around the homestead, slaughtering practices, milk consumption practices, sheltering livestock, and assisting them with difficult births. For household level risk factors such as seeing animals around the home, children's data from available parents were merged.

**Stage 3: Nested case-control.** A nested case-control study was carried out in February and March 2021. Since livestock ownership was low in the cohort, the nested case-control study, focused on community animal exposure, consumption, and sourcing of ASFs. RVFV seropositive participants were matched 1:4 to seronegative participants according to gender, and age ranges that reflected school age and general lifestyle patterns (0–8, 9–13, 14–19, 20–25, 26–35, 36–45, 46–60, 60+ years). Age and gender were prioritized for matching, and, when possible, participants in the same or adjacent study zone were selected as negative controls. For the seropositive children and their respective negative controls, a proxy parent or other adult in the household answered the questionnaire with additional questions for child exposure. Again, where appropriate for household level risk factors, adult data were merged with their respective children. In addition, the survey included open-ended sections on knowledge of RVFV transmission and perceptions towards vaccination.

## Questionnaire administration

All data were collected by long-term trained personnel and administered in the participants' preferred language (English, Kiswahili, or local languages Digo (coast) and Luo (west)). Data were captured electronically, and a Redcap database was used for secure storage (http://redcap. stanford.edu).

## Laboratory analysis

All samples from enrollment in the parent study were tested for anti-RVFV, anti-CHIKV, and anti-DENV IgG antibodies using in-house standardized ELISA tests. These standardized indirect in-house ELISA methods have been conducted previously from our study sites and specific methods for RVFV (4,25), CHIKV, and DENV (26,27) are described elsewhere.

Briefly, for RVFV ELISA, MP-12 antigen was coated on 96-well plates, clinical samples were diluted 1:10 and added to each plate followed by an anti-goat secondary antibody. ODs were read at 405 nm after 30 minutes incubation, and a strict cutoff was set for OD that was 2x the positive controls.

## Statistical methods

The primary outcome for all analysis was the individual participant's binary anti-RVFV IgG ELISA result (positive or negative). Summary descriptive statistics were computed to assess

demographic variables, and exposures were compared between seropositive and seronegative participants. Bivariate analyses of demographics, and animal exposures were conducted using Fisher's exact tests for categorical and binary predictors, and independent t-tests were used for continuous predictors such as age. All statistical analyses were conducted using R, and R Studio (Version 1.3.1093). For the follow-up data, we computed bivariable logistic regressions and decrease the probability of a type 1 error in assessing significance of the variables from this large dataset, we performed a Bonferroni adjustment (0.05/20) and set the threshold for significance at 0.0025 when interpreting significance for this portion of the analysis.

Multivariable models were built and included data from all participants that answered the follow-up questions. A Pearson correlation coefficient was computed to assess the linear relationship between related variables and dependent variables that had r values greater than 0.4 or less than -0.4 were considered were considered strongly correlated and correlation was considered significant at the 0.05 level (two-tailed). For significant strong correlation between two variables, one variable was dropped from initial inclusion in the models based on contextual understanding of the parameters. Models were initially built using variables that were significantly associated with RVFV in the bivariate analysis and biologically plausible variables approaching significance with a cutoff p-value of 0.1. Logistic models were also built separately for each study site to determine location specific predictors of exposure. Modeling was performed with the glmer and lme4 packages in R studio, and unconditional logistic regression with stepwise backward elimination was used to obtain the final three predictive models. Biologically plausible variables that were significant in univariate analysis were kept in models when removing variables stepwise. Focus was directed on significance of predictors based on the Wald test p-value. The Akaike Information Criterion (AIC) was recorded to monitor model quality as insignificant variables were removed.

For case-control data, odds ratios (OR) and 95% confidence intervals (CI 95%) were calculated to compare differences between RVFV exposed and unexposed participants using the epiR package. The case-control data set had a limited number of comparisons and thus a significance level of 0.05 was set as the threshold for this analysis. Exposure variables were categorized as individual risk factors of household level risk factors, and parent proxies were included in the analysis as deemed appropriate. For RVFV knowledge, risk assessment, and household level practices, all participants were included in the analysis. Where appropriate, open-ended questions were thematically analyzed, and data were translated to binary or categorical variables.

## Results

Total participants at each stage of sampling are summarized in Fig 2. The high rate of loss to follow-up (42%, 1,495/3,560) in the parent study and consequently lack of availability to respond to the animal exposure risk questionnaire is, at least in part, due to the study's pause from March 2020 to July 2020 during the national COVID-19 pandemic lockdown when many participants reported to have returned to their rural homes. This effect was more profound at the larger urban site in the west, Kisumu (51%, 868/1,687), compared to Ukunda on the coast (33%, 627/1,873) (Fig 2).

### Stage 1: Baseline demographics and enrollment

A summary of demographics is presented in Table 1. Anti-RVFV IgG antibodies were detected in 57/3,560 participants (1.6% seroprevalence), with the west study site having a slightly higher seroprevalence (1.8%) than the coast site (1.5%) ($p = 0.08$). Overall, older males had higher rates of exposure to RVFV ($p = 0.02$). In Kisumu, age group 5–17 years had higher

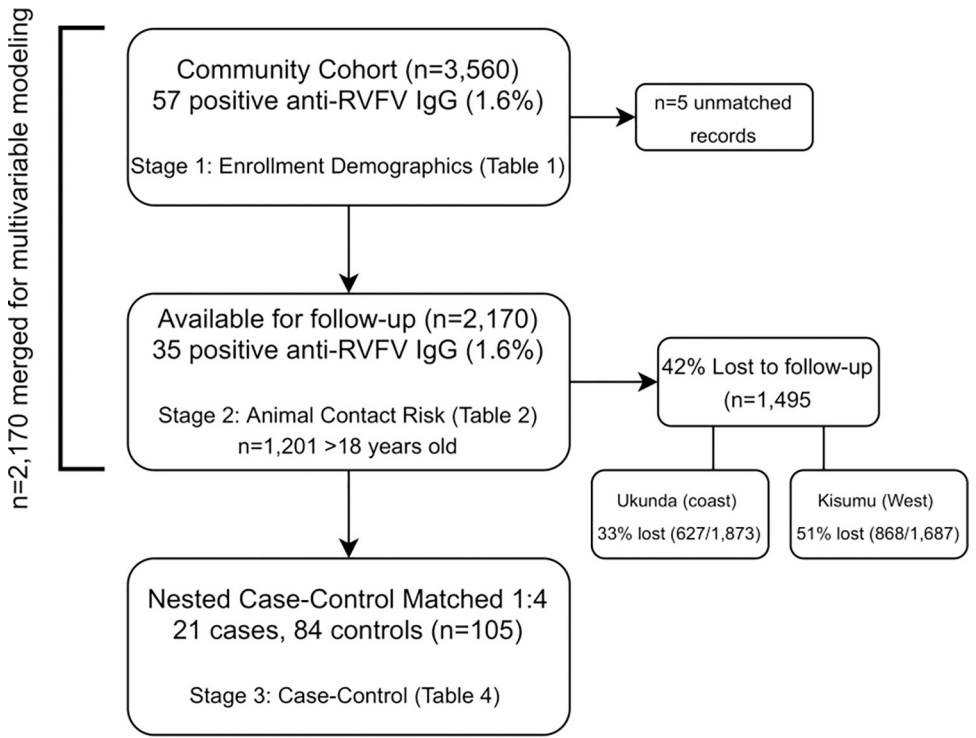

**Fig 2. Study design indicating total number of participants assessed at each stage.**

seroprevalence (2.1%) than everyone greater than 18 years (1.7%) (*p = 0.02*). In Ukunda, no participants younger than 5 years old were seropositive. (Fig 3). The lack of piped water in the home (*p = 0.002*) and use of small water jugs (15–20 liters) on a cart for the household water supply, was associated with RVFV exposure (*p<0.0001*).

## Exposure to other arboviruses

Of the 57 RVFV seropositive participants, 88% (50/57) had antibodies indicating prior exposure to at least one other arbovirus, with 51% (29/57) participants having exposure to all three arboviruses (RVFV, DENV, CHIKV). The west (Kisumu) had more participants with RVFV and CHIKV antibodies and the coast (Ukunda) had more participants with RVFV and DENV antibodies (Fig 4). For the 14 participants exposed to all three arboviruses in the west, 50% (7/14) were less than 14 years old. In fact, of all RVFV seropositive children, younger than 14 years old included in this study (n = 17), 79% (11/14) had been exposed to all three arboviruses at any time over their lifetime.

## Stage 2: Animal-exposure risk factors

The results of animal contact for individuals >18 years old (n = 1,201) and households (n = 2,170) are summarized in Table 2.

Overall, animal ownership was very low in our community cohort: 9% poultry, 6% dairy cows, 1.4% dogs, 3% cats, 1% goats, 0.4% beef cattle 0.2% sheep, and no pigs. None of the study participants reported seeing any wild ruminants around their homes and none of the RVFV seropositive participants (n = 35) reported owning ruminants at their urban home. Only 11% of adult individuals (130/1,201) of people provide care for animals that do not belong to them including 2/24 of seropositive that fed roaming animals or helped provide

**Table 1. Participant demographics overall and for RVFV seropositive participants.**

| | Variable | N = Overall Cohort (%) | Anti-RVFV IgG+ (%) | % Seropositive (IgG+/Total) | P-value* |
|---|---|---|---|---|---|
| **Total** | | **3,560 (1.6)** | **57** | **1.6** | |
| **Gender** | Male | 1,334 (37) | 32 (56) | 2.4 | **0.02** |
| | Female | 2,226 (63) | 25 (44) | 1.1 | |
| **Age (Continuous)** | Median [min-max] [1st Qu., 3rd Qu.] | 21 [1–85] [9,34] | 24 [3–66] [13,36] | | **0.04** |
| **Age Groups** | < 5 years | 396 (11) | 2 (3.5) | 0.5 | 0.25 |
| | 5–17 years | 1,156 (32) | 17 (30) | 1.5 | |
| | 18+ year | 2,008 (56) | 38 (67) | 1.9 | |
| **Location** | Coast (Ukunda) | 1,873 (53) | 27 (47) | 1.4 | 0.08 |
| | West (Kisumu) | 1,687 (47) | 30 (53) | 1.8 | |
| **Type of housing zone** | Unplanned | 1,917 (54) | 27 (47) | 1.4 | 0.73 |
| | Unplanned/Intermediate | 918 (26) | 18 (32) | 2.0 | |
| | Planned | 725 (20) | 12 (21) | 1.7 | |
| **Level of education** | No education | 361 (10) | 2 (4) | 0.6 | 0.22 |
| | Some primary school | 1,110 (31) | 16 (28) | 1.4 | |
| | Completed primary school | 384 (11) | 12 (21) | 3.1 | |
| | Some secondary school | 302 (9) | 1 (2) | 0.3 | |
| | Completed secondary school | 593 (17) | 12 (21) | 2.0 | |
| | College or University | 344 (10) | 7 (12) | 2.0 | |
| **Piped water in home or yard** | Yes, includes shared tap | 3100 (87) | 43 (75) | 1.4 | **0.002** |
| | No (jugs on cart or well) | 460 (13) | 14(25) | 3.0 | |
| **Alternative water source** | Small jugs on cart [1] | 78 (2) | 6 (11) | 7.7 | **<0.0001** |
| | All other (including piped) | 3,452 (98) | 51 (89) | 1.5 | |
| Formal occupation[2] (only adults) | Yes | 1,098/1,979 (55) | 26/37 (70) | 2.4 | 0.09 |
| | No | 881/1,979 (45) | 11/37 (30) | 1.2 | |
| Traveled in the past 6 months[3] | Yes, traveled anywhere | 649/3,261 (20) | 10/42 (24) | 1.5 | 0.80 |
| **Sleep under a mosquito net** | Yes | 576/2,641 (22) | 12/40 (30) | 2.1 | 0.33 |

[1] Plastic jugs size 15–20 liters filled at another location and transported on a cart to the household

[2] Formal occupation was self-defined as a consistent way to make money for oneself or the household

[3]Defined as sleeping outside of the original residence for more than 1 night

*P value calculated with Chi-square test in epiR 2by2 package, significance level of 0.05

Abbreviations: Qu: quartiles

medical care for neighborhood animals. None of the study participants herded animals that do not belong to them and none of the RVFV seropositive participants reported to slaughter animals, house them at night, or assisted livestock birth. However, RVFV seropositive participants were more likely to see goats (OR = 1.9 [CI 95%: 0.95–4.07], *p = 0.05*) and poultry (OR = 2.0 [CI 95%: 0.96–4.23], *p = 0.05*) around their homes.

*For consumption of animal products*, 35% (420/1,201) of the adults that responded individually drank milk. Only ten received their milk from a personal dairy animal and none of these participants were RVFV seropositive. Kisumu (8%) had a higher rate of raw milk drinkers compared to Ukunda (4%) (*p = 0.06*) (Table 2). These results were explored further in the nested case-control reported below.

## Multivariable modeling

The results of multivariate modeling are summarized in Fig 5. Not boiling milk and consuming raw milk were weakly positively correlated according to Pearson's correlation test (r

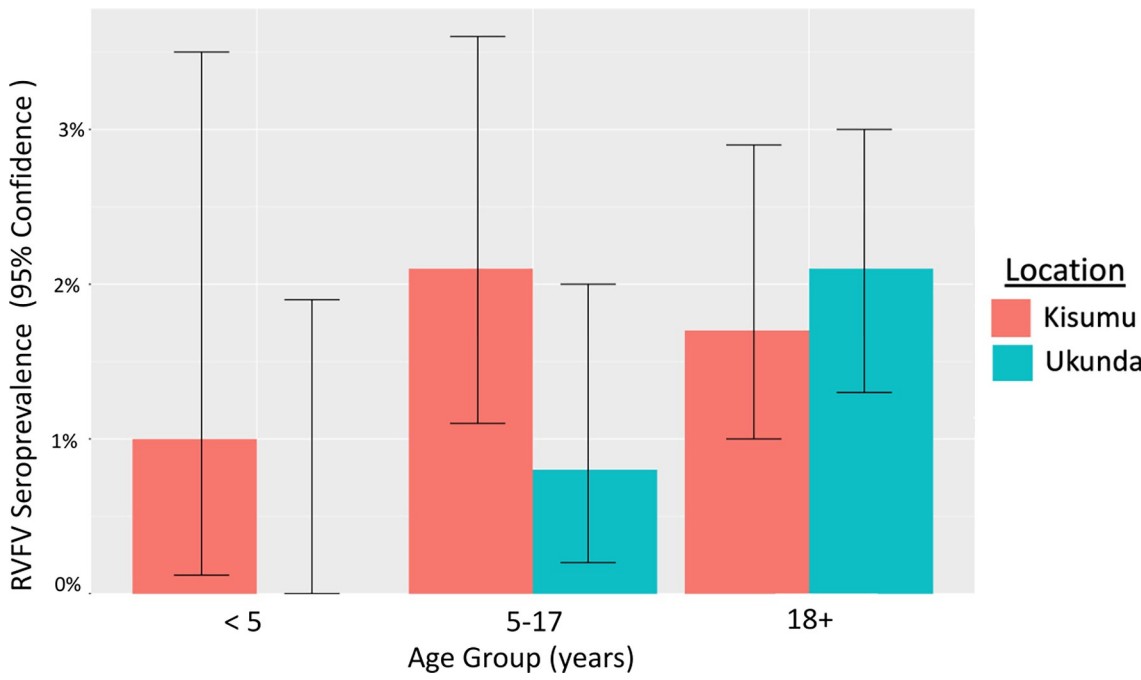

**Fig 3. RVFV IgG seropositivity by age group and location.**

(2,168) = 0.22, $p<0.002$). Given the strength of this association, both variables were included in the initial models. Another weak positive correlation between participant age and consumption of raw milk was identified (r (2,168) = 0.15, $p<0.002$) and again, both variables were

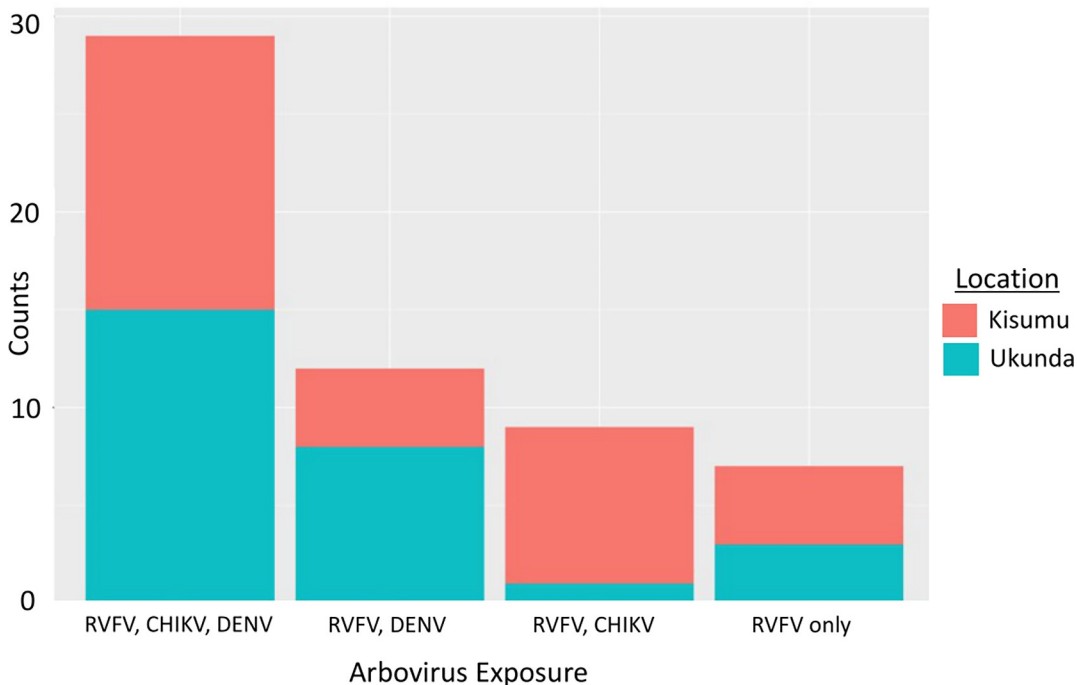

**Fig 4. Counts of RVFV seropositive participant's (n = 57) co-exposure to other arboviruses.**

**Table 2. Animal contact bivariate analysis for Rift Valley fever virus exposure.**

| Risk Factor | | | N = (% of total) | RVFV + (% Sero-positive) | OR (CI 95%) | P value |
|---|---|---|---|---|---|---|
| *Household level risk n = 2,170* | | | | *RVFV+ n = 35* | | |
| Seeing animals around home | | | | | | |
| | Dairy cattle | | 404 (19) | 11 (31) | 2.0 (0.88–4.34) | 0.06 |
| | Beef cattle | | 489 (23) | 12 (34) | 1.8 (0.82–3.83) | 0.09 |
| | Goats | | 767 (35) | 18 (51) | 1.9 (0.95–4.07) | 0.05 |
| | Sheep | | 106 (5) | 0 | 0 | |
| | Poultry | | 873 (40) | 20 (57) | 2.0(0.96–4.23) | 0.04 |
| Animals entering the home | Yes, any animal | | 248 (11) | 6 (17) | 1.3(0.42–3.47) | 0.58 |
| Consumption of raw milk in the household | Yes | | 79 (7) | 3 (9) | 2.5(.49–8.40) | 0.13 |
| | Kisumu (n = 916) | Yes | 25 (3) | 2 (6) | 5.9(0.61–28.25) | 0.02 |
| | Ukunda (n = 1,254) | Yes | 54 (4) | 1 (3) | 1.2(0.03–7.66) | 0.87 |
| *Individual level risk n = 1,201* | | | | *(RVFV+ n = 24)* | | |
| Providing care animals not owned | Yes | | 130 (11) | 2 (8) | 0.7(0.08–3.09) | 0.69 |
| | No | | 1,071 (89) | 22 (92) | | |
| Assisted livestock birth in past one month | Yes | | 16 (1) | 1 (4) | 3.3(.08–23.88) | 0.24 |
| | No | | 1,185 (99) | 23 (96) | | |
| **Slaughtered a ruminant past one month** | Yes | | 0 | 0 | | |
| | No | | 1,201 (100) | 24 (100) | | |

P value: Calculated with glm logistic regressions; p >0.002 significant

retained in the initial model. However, variables to determine water source, the availability of piped water and the use of a small water jugs, were highly negatively correlated (r (3,558) = -0.43, $p<0.002$) and the use of small jugs having the more significant p-value in bivariate analysis was chosen for inclusion in model building.

| | Overall Model | | | Kisumu Model | | | Ukunda Model | | |
|---|---|---|---|---|---|---|---|---|---|
| *Predictor* | *Odds Ratios* | *CI 95%* | *p* | *Odds Ratios* | *CI 95%* | *p* | *Odds Ratios* | *CI 95%* | *p* |
| Intercept | 0.05 | 0.02 – 0.14 | **<0.001** | 0.01 | 0.01 – 0.02 | **<0.001** | 0.04 | 0.01 – 0.17 | **<0.001** |
| Small Jugs for Water Source | 5.36 | 1.23 – 16.44 | **0.009** | 4.88 | 1.07 – 16.31 | **0.018** | | | |
| See Goats Around Home | 2.34 | 1.18 – 4.69 | **0.015** | | | | 2.97 | 1.18 – 8.47 | **0.027** |
| Gender | 0.36 | 0.18 – 0.71 | **0.004** | | | | 0.36 | 0.14 – 0.88 | **0.027** |
| Consume Raw Milk | | | | 6.28 | 0.94 – 25.21 | **0.021** | | | |
| Observations | 2170 | | | 916 | | | 1254 | | |
| $R^2$ Tjur | 0.011 | | | 0.013 | | | 0.010 | | |

**Fig 5. Multivariate models to predict RVFV seropositivity overall and by each location.**

The overall model (n = 2,170) showed household use of small jugs as the water source, seeing goats around the home, and gender were significant predictors of individual RVFV seropositivity. However, when models were built for Kisumu (n = 916) and Ukunda (n = 1,254) separately, local predictors of disease were revealed. In Kisumu, the use of small jugs (15–20 liters) as the primary household water source and consumption of raw milk were significant predictors of RVFV seropositivity. In this model, boiling milk was not protective for RVFV exposure. Both significant factors for Kisumu were not significant in the Ukunda model, which instead showed that seeing goats around the home and gender as significant predictors. Since seeing goats and consumption of raw milk were biologically plausible predictors, they were left in the models when removing non-significant variables one by one, and ultimately were significant while the AIC remained stable.

## Stage 3 results: Nested case-control

The odds ratios of the nested case-control study are summarized in Table 3 and the descriptive statistics are outlined in the following sections. None of the participants previously knew they had been exposed to RVFV before and none of the participants had been tested for RVFV prior to this study. 19 participants overall, including three cases, knew how RVFV was spread although due to the study design, this could be attributed to the brief introduction given during enrollment in the parent study. All 105 participants received this same briefing at the time

**Table 3. Odds ratios to compare community animal exposures between cases and controls.**

| | Variable | | Case | Control | OR [CI 95%] | P-value* |
|---|---|---|---|---|---|---|
| **Consumption** | Milk source | Official | 10 | 46 | 3.3 [0.86–12.5] | 0.07 |
| | | Unofficial | 5 | 7 | | |
| | Purchase from milk Vendors | Yes | 4 | 5 | 3.5 [0.8–15.2] | 0.08 |
| | | No | 11 | 48 | | |
| | Purchase milk from dairy cow owners | Yes | 5 | 15 | 1.4 [0.40–4.89] | 0.59 |
| | | No | 9 | 38 | | |
| | Consumption of homemade yogurt[1] | Yes | 3 | 8 | 1.0 [0.24–4.28] | 0.98 |
| | | No | 17 | 46 | | |
| | Handle ruminant Blood[1] | Yes | 3 | 5 | 3.0 [0.6–14.4] | 0.16 |
| | | No | 11 | 55 | | |
| **Community animals** | Touch animals not owned[1] | Yes | 3 | 2 | 7.3 [1.1–48.2] | **0.02** |
| | | No | 12 | 58 | | |
| **Health** | Intermittent blurry vision[1] | Yes | 6 | 4 | 9.3 [2.2–39.7] | **<0.001** |
| | | No | 9 | 56 | | |
| **Ruminant ownership in rural home[2]** | Own ruminants | Yes | 6 | 13 | 2.2 [0.7–6.7] | 0.16 |
| | | No | 13 | 71 | | |
| | Own beef cattle | Yes | 4 | 3 | 6.7 [0.8–56.2] | 0.07 |
| | | No | 2 | 10 | | |
| | Graze with wild animals[2] | Yes | 2 | 2 | 9.0 [0.5–155.2] | 0.10 |
| | | No | 9 | 2 | | |
| | Graze with others[2] | Yes | 2 | 3 | 5.3 [0.3–82.8] | 0.21 |
| | | No | 1 | 8 | | |

[1]Personal risk, excluded proxy parents

[2]n = 19 participants whose family in their rural homes owned ruminants

*P value calculated with Chi-square test in epiR 2by2 package, significance level of 0.05

of the case-control study, 18 (five cases) perceived they were at risk of RVFV infection giving reasons that included exposure to animals and daily mosquito bites. For the 19 participants (six cases) that remembered RVFV but did not feel like they were at risk gave reasons such as: not owning or raising animals (n = 4, two cases), the belief that it was an animal only disease (n = 4, zero cases), reasons related to their geographical location (n = 3, zero cases), and factors related to how they consume animal products (n = 2, zero cases).

RVFV seropositive participants were nine times more likely to sometimes experience a sudden decrease in vision compared to RVFV seronegative participants (OR = 9.3 [CI 95%: 2.19–39.7], *p = <0.001*). For the seven seropositive participants that sometimes had decreased vision, most of them reported this as a recent phenomenon.

### Indirect animal exposures: Community and consumption

To assess community animal exposures, case control participants were asked if they touch animals that do not belong to them and the odds of RVFV seropositive participants were seven times higher for touching animals that didn't belong to them than seronegative participants (OR = 7.3 [CI 95%: 1.1–49.2], *p = 0.02*), giving reasons that included chasing animals away from their home, offering water or food scraps, and slaughtering for festivals (one case). None of the participants in this study knew how to recognize RVF disease in livestock.

For consumption related risks, we asked participants where they purchased milk from, and 9 (4 cases) purchased from a milk vendor (OR = 3.5 [CI 95%: 0.8–15.2], *p = 0.08*). When all milk sources from the unofficial value chain were grouped (including direct purchases from dairy animal owners and milk vendors) and compared to official sources (formal commercial packaging), 11 participants (5 cases) purchased milk from unofficial sources compared to official (OR = 3.9 [CI 95%: 1.0–15.4], *p = 0.04*). There were 20 (five cases) participants who purchased milk from dairy owners directly and two justified their decision not to boil milk themselves because they assumed that the person selling them the milk had already boiled it.

Our open-ended questions revealed cultural practices that persist in the urban setting including two total observations of tasting beef before cooking it and both were RVFV seropositive participants. Four RVFV-unexposed participants reported adding raw stomach content to grilled meat. 10 participants (three cases) reported adding abnormal raw milk to their cooked vegetables and 12 participants reported handling animal blood during food preparation, though there was also no difference between RVFV exposed and unexposed participants.

### Child exposures to RVFV

Participants less than 18 years old were considered children in this study. When parents of seropositive children responded to questions about their child's exposure, only one seropositive child provided care for dairy animals that included milking. The seropositive children followed a similar trend to the adults as four out of six routinely travel with their family to their countryside homes, mostly in western Kenya. Children did not herd animals or care for them outside of the home such as at school.

### RVFV vaccination perceptions

91% (97/105) of participants reported that they would be willing to accept a future RVFV vaccination for humans. Of the eight participants that would refuse, one participant was seropositive for RVFV, and they reasoned that they would not accept the vaccination unless they had the disease. Others did not like injections (n = 2), had not heard of the disease before (n = 2) and said they did not own any ruminants (n = 2). In general, there was a high level of acceptance despite low overall initial knowledge of RVFV. Of the nine participants that said they

had previously declined their child vaccination for another infectious disease, four had RVFV exposure. Of the 75 participants that had children, 93% (70/75) of them would want their child to get vaccinated if RVFV arrived in their area.

For animal vaccinations, there were less health concerns and decisions were focused on information seeking rather than refusal. When participants were asked who should get vaccinated if a disease is shared between animals and humans, and 15 participants responded animals, six responded humans, 82 responded both, and three were unsure. One participant expressed concern that they are often unaware of what disease their animal is being vaccinated for and others (n = 3) feared that their livestock would die or not be safe to eat after receiving vaccination.

## Discussion

The risk factors we found in this RVFV urban exposure risk assessment differ from what has been previously described [5, 25, 27] and we have demonstrated exposure risks independent of livestock ownership. None of the RVFV seropositive participants in our study were herding, sheltering livestock inside the homes, or assisting with difficult births and livestock abortion, and only one slaughtered livestock. Given the availability of hosts, vectors, and continuous introduction of livestock and animal products from high-risk areas, urban areas in Kenya are indeed at risk for RVFV. However, understanding the extent of human risk requires a more inclusive definition and must consider community livestock exposures, sourcing of milk, and household level risk such as living around livestock and not having access to piped water.

Our study provides evidence that consumption of raw milk and potentially handling of raw milk may be risk factors for RVFV exposure in the urban setting, which may arrive in the urban center from surrounding rural areas with higher risk of RVFV. The true infectious potential for raw milk consumption and aerosolization of viral particles remains unknown. If raw milk can serve as a means for viral spread, we caution the assumption that urban areas have a low risk for RVFV introduction as urban areas have an increased demand for ASFs and fewer small-scale farmers, so they import animals and animal products from a wide geographical area. However, our study did not determine if peoples' raw milk consumption occurs while they are in the urban center or when they return to their family's rural home and practices may differ during rural home visits.

We also found differences between our two urban sites which highlights that although urban areas share commonalities, risk is still dependent on local practices and cultural norms, even within the same country. In Kenya, most of the milk is produced in the Rift Valley region which supplies the urban Kisumu informal market, which may explain why consumption of raw milk was only a risk factor in the west [34]. The higher seroprevalence in children aged 5–17 (2.1% compared to 1.6% in adults) in Kisumu is concerning and we did not find any direct epidemiological links to explain this finding. In general, children often consume milk at higher rates than adults and children should not be excluded from any prevention and control measures for RVFV. On the other hand, the county where Ukunda is located, Kwale county, had previously been classified as high-risk for epizootics [33] and seeing goats around the home was linked to RVFV seroprevalence and could indicate that RVFV has already established a low-level urban transmission cycle in smaller urban coastal towns. Goats are one of the animal species highly susceptible to RVFV [1] and could be acting as viral amplifiers. Their presence could attract mosquito vectors via water containers around the household rather than deter them as with the Anopheles spp. and cattle for malaria zooprophylaxis [35]. RVFV associations with water source and unsafe storage of water may signal ongoing local interepidemic vector transmission. Larval source reduction practices could decrease the risk of urban arbovirus exposure [29, 36], including RVFV.

As this was the first major assessment of urban community risk factors for RVFV exposure, our study had several limitations. Given the profound inequalities often present in urban centers, our results could be confounded by socioeconomic features not assessed and future studies should consider conducting in-depth wealth assessments. Likely the greatest limitation in interpreting our results and perhaps in the assessment of urban populations in general, is the highly mobile and dynamic lifestyle patterns of urban people. As anti-RVFV antibodies are thought to be lifelong, we cannot conclude that the urban populations studied were exposed to RVFV in their urban town; they may have been exposed before they relocated to the urban area as the sharp increase in urbanization has occurred over the lifetime of many of our participants or during travel to their rural homes. We collected information on travel outside of the urban setting to control for this possibility and 80% of RVFV exposed individuals had no recent outside travel (Table 1) which could indicate urban transmission potential, although we cannot collect lifetime travel information, and travel could have indeed been the source of infection for our urban participants. The high loss to follow-up due to the Covid-19 pandemic pause have altered our overall seroprevalence estimates and subsequent lack of availability for follow-up may have affected our case control study. Additionally, as plaque reduction neutralization assay is the gold standard for anti-RVFV IgG identification, we consider cross-reactivity of our in-house ELISA assay to be a limitation of this study. Other related bunyaviruses have been detected in Kenya [37–39], however, RVFV remains the most prevalent. This study may have little generalizability to other urban populations as we have only included two urban centers with varying populations; however, as with the vector transmission pathway, this diversity may indeed provide an advantage to disentangle risk factors for consuming animal products from other basic livestock rearing activities.

Our study included an initial assessment of human vaccination perceptions in an urban area for a near future RVFV vaccines. Overall, vaccine acceptance was high in these urban communities although challenges around understanding of zoonosis may thwart attempts to use vaccination as a preventive strategy or outbreak control measure. Vaccinating humans for RVFV in urban populations will be challenging without fully understanding who is at higher risk especially in the urban setting, where handling of ASFs is common and daily vector exposure is nearly unanimous. The overall lack of understanding of RVF disease transmission and risk will undoubtedly be connected to peoples' decision to accept a vaccination for themselves or their children [40]. Furthermore, disease ecology of an urban RVFV outbreak could be expected to have different risk factors for transmission. Importantly, we have highlighted the potential for RVFV exposure independent of livestock ownership which represents a major unqualified global risk. There is an urgent need to better qualify risk for urban inhabitants that live adjacent to high-risk regions and fully disentangle consumption related risks from other activities associated with livestock.

## Conclusion

In conclusion, consumption of raw milk, living around goats, and unreliable water sources are potential RVFV risk factors independent of livestock ownership in the urban setting in Kenya. RVFV risk without livestock ownership highlights a potentially greater importance of transmission from animal product consumption or directly from vector bites in urban settings of endemic countries. Active urban surveillance and active case finding are needed in urban areas when rural outbreaks occur nearby. Although overall risk may be lower than in rural pastoralist areas, RVFV risk is present and should be further investigated to prepare mitigation efforts. More studies on RVFV infectivity in animal-source food handling and consumption are needed to assess this more widespread risk. We must understand the infectivity of these

highly mobile animal products and specific points at which human risk is high so that prevention campaigns in urban areas can be better targeted. In addition, there may be additional vaccine hesitancy for RVFV present in the urban areas and more research and awareness building are critical to reap the full benefit of human vaccination before human vaccines are available. RVFV must not be seen as a disease concern only in pastoralist communities in rural areas; urban areas must prepare and deploy surveillance efforts when RVFV outbreaks are detected in endemic regions.

## Supporting information

**S1 Appendix. Questionnaires from parent study.**
(PDF)

**S1 File. PLOS global public health inclusivity questionnaire.**
(DOCX)

## Acknowledgments

We thank the local communities, health officers and administrators in the study sites in Kisumu and Ukunda for allowing us to work in their areas of their jurisdiction. We also thank all project staff from the larger R-01 study that collected data and safely restarted fieldwork after COVID-19 lockdowns were lifted. In addition, we thank Olivia Paige and Maria Hernandez with the GHES Fellowship for their support.

## Author Contributions

**Conceptualization:** Keli Nicole Gerken, Francis Maluki Mutuku, Bryson Alberto Ndenga, A. Desiree LaBeaud.

**Data curation:** Keli Nicole Gerken, Gladys Adhiambo Agola, Said Malumbo, Karren Nyumbile Shaita.

**Formal analysis:** Keli Nicole Gerken, Gladys Adhiambo Agola, Eduardo Palacios Fabre.

**Funding acquisition:** Keli Nicole Gerken, A. Desiree LaBeaud.

**Investigation:** Keli Nicole Gerken, Francis Maluki Mutuku, Bryson Alberto Ndenga, A. Desiree LaBeaud.

**Methodology:** Keli Nicole Gerken, Francis Maluki Mutuku, Bryson Alberto Ndenga, Eleonora Migliore, Izabela Mauricio Rezende, A. Desiree LaBeaud.

**Project administration:** Keli Nicole Gerken, Francis Maluki Mutuku, Bryson Alberto Ndenga, A. Desiree LaBeaud.

**Resources:** Francis Maluki Mutuku, Bryson Alberto Ndenga, Eleonora Migliore, A. Desiree LaBeaud.

**Software:** A. Desiree LaBeaud.

**Supervision:** Francis Maluki Mutuku, Bryson Alberto Ndenga, A. Desiree LaBeaud.

**Validation:** Keli Nicole Gerken, Gladys Adhiambo Agola, Eduardo Palacios Fabre.

**Visualization:** Keli Nicole Gerken, Gladys Adhiambo Agola.

**Writing – original draft:** Keli Nicole Gerken.

**Writing – review & editing:** Keli Nicole Gerken, Francis Maluki Mutuku, Bryson Alberto Ndenga, Gladys Adhiambo Agola, Eduardo Palacios Fabre, Izabela Mauricio Rezende, A. Desiree LaBeaud.

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
