## [Decision Letter · Decision Letter 0]

18 Apr 2022

PGPH-D-22-00097

Urban risk factors for human Rift Valley fever virus exposure in Kenya

Dear Dr. Gerken,

Thank you for submitting your manuscript to PLOS Global Public Health. After careful consideration, we feel that it has merit but does not fully meet PLOS Global Public Health’s publication criteria as it currently stands. Therefore, we invite you to submit a revised version of the manuscript that addresses the points raised during the review process.

Please address all the comments and queries raised by the reviewers ( below). 

We look forward to receiving your revised manuscript.

Kind regards,

Nusrat Homaira

Academic Editor

Journal Requirements:

2. Your co-authors:

Francis Maluki Mutuku -fmutuku73@gmail.com

Eleonora Migliore -elemig16@stanford.edu

Eduardo Fabre -edupalacios95@gmail.com

Said Malumbo -saidmalumbo@yahoo.com

Karren Nyumbile Shaita -patiencekarren49@gmail.com

Izabela Mauricio Rezende -irezende@stanford.edu

,have not confirmed authorship of the manuscript. We have resent them the authorship confirmation email; however please check that the above email address for them is correct and follow up personally to ensure they confirm. 

Please note that we cannot proceed your manuscript  until we have received confirmations from all co-authors.

3. Please amend your detailed Financial Disclosure statement. This is published with the article, therefore should be completed in full sentences and contain the exact wording you wish to be published.

iii). State what role the funders took in the study. If the funders had no role in your study, please state: “The funders had no role in study design, data collection and analysis, decision to publish, or preparation of the manuscript.”

4. Please provide  separate figure files in .tif or .eps format only and remove any figures embedded in your manuscript file.  Please ensure that all files are under our size limit of 20MB.  

For more information about how to convert your figure files please see our guidelines: Once you've converted your files to .tif or .eps, please also make sure that your figures meet our format requirements

5. Please provide us with a direct link to the base layer of the map used in Fig 1 and ensure this location is also included in the figure legend. 

Please note that, because all PLOS articles are published under a CC BY license (creativecommons.org/licenses/by/4.0/), we cannot publish proprietary maps such as Google Maps, Mapquest or other copyrighted maps. If your map was obtained from a copyrighted source please amend the figure so that the base map used is from an openly available source.

Please note that only the following CC BY licences are compatible with PLOS licence: CC BY 4.0, CC BY 2.0  and CC BY 3.0, meanwhile such licences as CC BY-ND 3.0 and others are not compatible due to additional restrictions. If you are unsure whether you can use a map or not, please do reach out and we will be able to help you. 

The following websites are good examples of where you can source open access or public domain maps:

Additional Editor Comments (if provided):

Reviewers' comments:

Reviewer's Responses to Questions

**Comments to the Author**

1. Does this manuscript meet PLOS Global Public Health’s publication criteria? Is the manuscript technically sound, and do the data support the conclusions? The manuscript must describe methodologically and ethically rigorous research with conclusions that are appropriately drawn based on the data presented.

Reviewer #1: Yes

Reviewer #2: Partly

2. Has the statistical analysis been performed appropriately and rigorously?

Reviewer #1: Yes

Reviewer #2: Yes

3. Have the authors made all data underlying the findings in their manuscript fully available (please refer to the Data Availability Statement at the start of the manuscript PDF file)?

Reviewer #1: Yes

Reviewer #2: Yes

4. Is the manuscript presented in an intelligible fashion and written in standard English?

Reviewer #1: Yes

Reviewer #2: Yes

5. Review Comments to the Author

Reviewer #1: The manuscript discusses the exposures to Rift Valley fever in urban settings of African country that experience regular epidemics. This aspect of Rift Valley fever has not been researched much and this manuscript is therefore novel. I definitely recommend publication of this manuscript after revision of some items of the analysis. I have provided an annotated copy of your manuscript with my comments and suggestions. Overall, this is a good research study and worthwhile for publication.

Reviewer #2: General Comment: Rift Valley fever is indeed of great concern, and it is critical that we obtain a better understanding of the numerous risk factors that might promote outbreaks of RVF. This study is a good start, but I have indicated a number of questions below. One additional comment, yes, there was an association between unreliable water sources and the presence of RVFV antibody prevalence. However, was this due to the unreliable water, or more likely that the unreliable water source was linked to the actual factors (possibly standard of living related) that were the real cause of the higher prevalence. The authors inconsistently used numerals and written numbers for numbers less than 10. I indicated a number of these, but there are others. This indicates a level of carelessness and needs to be checked more carefully.

Specific Comments:

1. Line 81: Yes, many mammal species tested do appear to replicate RVFV. However, does RVFV replicate in any avian species? You might want to change “animals,” which includes birds, to “mammals.”

2. Line 101: Yes, I know that Kisumu and Ukunda are indicted in Figure 1, but why not modify this sentence to, “…two urban sites in Kenya, Kisumu and Ukunda, from December…?”

3. Line 132: Use of the word, “breed.” I know that breed is commonly used to refer to mosquito larval habitat, but breed has sexual implication and mosquitoes do not breed in water, their larvae develop in water. Shouldn’t this be changed to, “… where mosquitoes mostly develop (28)?”

4. Line 333, Table 4: As with comments xx and xx below, I have some questions about the numbers in Table 4. There appears to be 15 “cases’ for most of the variables, i.e., milk source, purchase from vendors, community animals, and health. Why are there data from only 14 cases about purchasing milk from dairy cow owners or for handling ruminant blood, or from 21 cases for adding spoiled milk to cooked vegetables? Note, if the 18 cases adding spoiled milk to their vegetables was really 12 cases (to have 15 total cases), then wouldn’t the OR be over 1.5? Minor, but capitalization appears to be at random, i.e., Milk is capitalized in some but not other variables. Similar comment for ruminant.

5. Line 343: Here it states that one person was involved in slaughtering animals, yet in table 2, it states than 0/1,201 of the participants slaughtered animals. Which is it?

6. Lines 368-369: On line 368 it states that 97/105 of the participants reported willingness to take the vaccine (i.e., 8 unwilling), but on the next line, it states that of the nine seropositive participants that would refuse… How could there be nine refusers? More importantly, if the second sentence is correct, then the refusal rate was 9/21 for the positive cases and 0/84 for the negative controls. That is amazingly significant, so much so that I think the sentence is wrong.

7. Lines 373-375: Is there a RVFV vaccine approved for human use in Kenya? If not, how could the nine participants have previously declined a vaccination for their child? Minor, but “nine seropositive” on lines 369 and “nine participants” on line 373 should be “9 seropositive” and “9 participants,” respectively.

Minor comments:

8. Lines 111-112: As “months” are a unit of measurement, shouldn’t this be “6 months after…?” See also line 141 where “two kilometers south…” should be 2 kilometers south…” or “2 km south…” Throughout the manuscript, I’m not certain if it should be “kilometers” or “km” and “meters” or “m.”

9. Line 161: Minor, but “RVFV exposed children” should be “RVFV-exposed children.”

10. Line 253, Figure 2: I am just curious as to why the number positive for RVFV IgG is formatted differently in the Community Cohort, “57 positive…(1.6%),” and the Available for Follow-Up, “(RVFV anti-RVFV IgG+ n=35).” Why not use the same format and use “35 positive anti-RVFV IgG (1.6%)?” Note, the positivity rate remained at 1.6%.

11. Line 263, Table 1: Why is the “0.02” for gender in bold? Yes, it indicates that gender is a significant predictor of the presence of RVFV antibodies, so I like it being highlighted, but what about the other significant ones? Either highlight all or none of them.

12. Line 279, Table 2: Are the numbers for consumption of raw milk correct? How could there be 79 total when there were 25 each in Kisumu and Ukunda? Similarly, if there were only 1,201 people, how could there be 16 who assisted a livestock birth and 1,192 who did not (16 + 1,192 = 1,208)?

13. Line 334: Shouldn’t “9 times more…” be “nine times more…?” Similar comment for line 336, and why is “Seropositive” capitalized?

14. Line 357: Shouldn’t “twelve participants…” be “12 participants…?” also, line 350. Shouldn’t (5 cases) be (five cases) and line 269 where 9 seropositive should be nine seropositive?

12. Line 395-396: Why write out “animal sourced foods” when “ASFs” has been established and used several times?

16. Lines 404-405: Was the seroprevalence, about 2%, in Kisumu significantly greater than that in Ukunda, about 1%? Yes, any seroprevalence is too high, but without giving any numbers in the discussion, simply stating, “The high seroprevalence in children aged 5-17 years in Kisumu is concerning” is a bit misleading.

17. Lines 412-413: Comments about poultry. First of all, as sort of mentioned, I do not believe that RVFV produces a viremia in poultry, and as such, they would not be involved in RVFV transmission. Rather than serve as an attractant for potential vectors, they are likely to reduce RVFV transmission, sort of like cattle and malaria, by drawing vectors away from other mammals that could produce a viremia and expand RVFV transmission.

18. Lines 422-423: Here the authors stated, “in the urban setting, where handling of raw meat and milk is common.” As far as I can tell, theis is the first mention of “raw meat” in the manuscript. Was there any difference in seroprevalence by consumption of raw meat? Second, according to Table 2, overall, <10% consumed raw milk. Is this really a “common” behavior?

19. Line 437: As the authors had just mentioned, “anti-RVFV antibodies are thought to be lifelong.” Thus, the fact that 80% of the RVFV exposed individuals had not had “recent” travel is misleading. Yes, the statement is meaningful, but I think that they need to qualify it better that “recent” travel may not have been the source.

20. References: These need to be formatted properly.

a. Only the first word and proper nouns in a reference title should be capitalized. See reference 2. Also, in reference 7, “fever” in Rift Valley fever should not be capitalized and in reference 12, “yellow” should not be capitalized as they are not proper nouns.

b. All proper nouns need to be capitalized. “Rift Valley fever” appears as “rift valley fever” in numerous references. See references 5, 6, 8, and others.

c. I believe that there should be a “space” after the “colon” after the volume number and before the page number.

d. I believe that all pages should be indicated, i.e., in reference 2, “PLoS Negl Trop Dis. 2014;8(9):17–9” should have been, “PLoS Negl Trop Dis. 2014;8(9): 17–19.” These need to be fixed for many of the references.

e. Shouldn’t the journal for reference 28 be “Parasit Vectors” instead of “Parasites and Vectors?”

6. PLOS authors have the option to publish the peer review history of their article (what does this mean?). If published, this will include your full peer review and any attached files.

**Do you want your identity to be public for this peer review?** For information about this choice, including consent withdrawal, please see our Privacy Policy.

Reviewer #1: No

Reviewer #2: No

---

## [Editor Report · Decision Letter 1]

8 Jun 2022

Urban risk factors for human Rift Valley fever virus exposure in Kenya

PGPH-D-22-00097R1

Dear Dr Gerken

We are pleased to inform you that your manuscript 'Urban risk factors for human Rift Valley fever virus exposure in Kenya' has been provisionally accepted for publication in PLOS Global Public Health.

Best regards,

Nusrat Homaira

Academic Editor